# Arbuscular Mycorrhizal Fungi Alter Plant and Soil C:N:P Stoichiometries Under Warming and Nitrogen Input in a Semiarid Meadow of China

**DOI:** 10.3390/ijerph16030397

**Published:** 2019-01-31

**Authors:** Linlin Mei, Xue Yang, Hongbing Cao, Tao Zhang, Jixun Guo

**Affiliations:** Institute of Grassland Science, Northeast Normal University, Key Laboratory of Vegetation Ecology, Ministry of Education, Changchun 130024, China; meill641@nenu.edu.cn (L.M.); yangx014@nenu.edu.cn (X.Y.); meilinlinmll@126.com (H.C.)

**Keywords:** arbuscular mycorrhizal fungi, global change, grassland ecosystem, stoichiometry, phosphorus limitation

## Abstract

Ecological stoichiometry has been widely used to determine how plant-soil systems respond to global change and to reveal which factors limit plant growth. Arbuscular mycorrhizal fungi (AMF) can increase plants’ uptake of nutrients such as nitrogen (N) and phosphorus (P), thereby altering plant and soil stoichiometries. To understand the regulatory effect of AMF feedback on plants and soil stoichiometry under global change, a microcosm experiment was conducted with warming and N input. The C_4_ grass *Setaria viridis*, C_3_ grass *Leymus chinensis*, and Chenopodiaceae species *Suaeda corniculata* were studied. The results showed that the mycorrhizal benefits for the C_4_ grass *S. viridis* were greater than those for the C_3_ grass *L. chinensis*, whereas for the Chenopodiaceae species *S. corniculata*, AMF symbiosis was antagonistic. Under N input and a combination of warming and N input, AMF significantly decreased the N:P ratios of all three species. Under N input, the soil N content and the N:P ratio were decreased significantly in the presence of AMF, whereas the soil C:N ratio was increased. These results showed that AMF can reduce the P limitation caused by N input and improve the efficiency of nutrient utilization, slow the negative influence of global change on plant growth, and promote grassland sustainability.

## 1. Introduction

Global climate change, including warming and increased atmospheric nitrogen deposition, has influenced the structure and function of terrestrial ecosystems over the last century [1]. It is predicted that global warming will continue to increase in the next 100 years [2]. Changes in temperature can affect plant productivity and community composition [3,4], e.g., by influencing nutrient demand and uptake [5]. Previous studies have shown that warming may alter carbon (C), nitrogen (N) and phosphorus (P) availability in soil [6] and nutrient cycles [7,8,9], simultaneously altering the stoichiometry of plants [4,10]. A previous study demonstrated that plant N and P contents declined because of warming, and plants invested fewer nutrients to produce proteins for sustaining biochemical reactions under warming conditions [11]. In addition, studies have shown that warming is likely to accelerate biological processes [12] and enhance decomposition and mineralization rates [13,14], which will enhance the supply rates of N and P to plants [11,15].

Anthropogenic N deposition, i.e., N deposition caused by human activities, is another important threat to ecosystem stability that reduces plant species richness and diversity, increases plant productivity and litter production [16], and affects biogeochemical cycles and net ecosystem C accumulation [17,18]. China is one of the most serious N deposition zones worldwide. Over the past 30 years, the average annual bulk deposition of N increased by approximately 80 kg ha^−1^ y^−1^ N [19]. N input decreases C mineralization, accelerates net N mineralization in soil [20,21], causes N:P imbalances in soil and indirectly affects plant growth [22]. Yue et al. [23] compiled a large number of studies to determine how plant and soil C:N:P stoichiometries respond to individual and combined effects of warming, N input and elevated CO_2_ concentrations. The results showed that the effects of N addition on plant C:N:P stoichiometry were stronger than those of warming. Most plants are N limited, but the increasing N input has significantly altered nutrient availability and has resulted in the transformation of N-limited to N and P colimited or P-limited ecosystems [24,25,26]. Our previous study [9] showed that the addition of N increased the amount of available N and the rate of N mineralization, but reduced soil available P; thus, the soil N:P ratio will decrease under climate change. A high P limitation in saline alkali soil is obvious in the Songnen meadow steppe. Therefore, exploration of the influence of warming and N input on plant and soil stoichiometries of grassland ecosystems, identification of a way to reduce the threat of grassland degradation caused by global change, and determination of a means to increase nutrient use efficiency will be beneficial to maintain sustainable grassland development and reduce environmental impacts.

Arbuscular mycorrhizal fungi (AMF) are among the most important microorganisms in terrestrial ecosystems [27,28]. AMF can form symbiotic relationships with the majority of land plants [29] and can increase plant N and P uptake and obtain C from the host plant, thereby altering the plant C:N:P stoichiometry [30]. Moreover, AMF enhance ecosystem stability and sustainability [31,32].

However, the development and species composition of AMF are often influenced by global change, such as warming and N input [33,34]. In our previous research [35], we found that AMF altered the species composition and productivity under warming and nitrogen (N) addition. However, the mechanism by which AMF affect plant community composition and productivity under warming and N addition is not well understood.

Here, we selected Setaria viridis (*L. Gramineae*) Leymus chinensis (Trin. Tzvel, Poaceae), and Suaeda corniculata (C. A. Mey, Chenopodiaceae), three typical dominant plants in the Songnen meadow steppe, to determine the effects of AMF on the C:N:P stoichiometry of plants and soils under global change. Chenopodiaceae species are highly tolerant to high salinity and drought stress, and an increasing number of studies have found that many chenopods can be colonized by AMF [36,37,38,39]. Our previous study showed that AMF increased the relative abundance and aboveground biomass of *S. viridis* and *L. chinensis*, but significantly reduced the relative abundance and aboveground biomass of *S. corniculata* under both warming and N addition [35].

An experiment was conducted using microcosms in a greenhouse. We assessed the total C, N and P contents of plants and soil. We hypothesized that (1) AMF improve C_4_
*Setaria viridis* (*L. Gramineae*) and C_3_
*Leymus chinensis* (Trin. Tzvel, Poaceae) N and P nutrition and decrease the N and P contents of Suaeda corniculata (C. A. Mey, Chenopodiaceae) under global change; (2) AMF have different contributions to the C:N:P stoichiometry of the three species under global change; and (3) AMF decrease the plant N:P ratio.

## 2. Materials and Methods 

### 2.1. Plant and Soil Preparation

We established experimental grassland microcosms with typical sodic saline meadow soil from the Songnen Meadow Ecological Research Station (44°45′ N, 123°45′ E), Northeast Normal University, Jilin Province, in northeastern China; this area has a semiarid temperate-zone monsoon climate and typical characteristics of a continental climate. The soil pH was 8.2. The vegetation of the experimental site is dominated primarily by *S. viridis*, *L. chinensis*, and *S. corniculata*. *S. viridis* is an indigenous C_4_ grass that associates strongly with AMF. *L. chinensis* is a C_3_ grass that gains little benefit from association with AMF [40,41], and *S. corniculata* is a member of the Chenopodiaceae family. The seeds of the three species and soil were collected from the Songnen meadow and were stored in a refrigerator at 4 °C before being used.

Topsoil (0–30 cm) was collected from the same site from which the plant seeds were collected. The soil was sieved (2 mm sieve) to remove large stones and plant roots and was sterilized twice using high-pressure steam at 121 °C for two hours each time to eliminate indigenous AMF.

We collected soil (500 g) from the Songnen meadow where warming and N input had been applied for five years and where Medicago sativa had been cultivated in 200 g of sterilized soil for four months. After the four month period, the aboveground biomass was removed, and the inoculum comprised spores, infected root fragments, hyphae, and soil (approximately 2000 spores per 50 g of soil).

### 2.2. Experimental Design

This experiment included four global change treatments: Control (CK), warming (W), N input (N), and a combination of warming and N input (WN). Each global change treatment included two AMF treatments: with AMF (‘AM’) and without AMF (‘NM’); each treatment was replicated four times, i.e., a total of 32 microcosms.

Sterilized soil was placed in microcosm pots (2.5 kg of dry soil per pot), and 200 g of AMF inoculum was added to produce the mycorrhizal (‘AM’) treatment. The same amount of sterilized soil (121 °C for two hours) was placed in the microcosm pots to produce the nonmycorrhizal (‘NM’) treatment. To ensure the rhizosphere microbial communities in the NM treatment were consistent with those in the AM treatment, a microbial suspension was prepared by sequential filtration of a soil extract obtained by orbital shaking (150 rpm) of a nonsterile soil, sterile dH_2_O (1:9 w/v) mixture for 30 min. The AM treatment received 10 ml of deionized water, and the NM treatment received 10 ml of filtrates that were free of mycorrhizal propagules [35].

Before planting, the seeds were surface disinfected with 10% (v/v) hydrogen peroxide for 5 min and washed five times with deionized water. The seeds were allowed to germinate at 20 °C for 48 h. Uniform seedlings were transplanted after 3 days into the grassland microcosms that had a height of 23 cm and a volume of approx. 2.5 kg (based on soil dry weight). The microcosms contained *Setaria viridis* (10 plants per pot), *L. chinensis* (15 plants per pot), and *S. corniculata* (5 plants per pot) [35].

The experiments were performed in phytotrons (LT/ACR-2002, E-Sheng Tech., Beijing, China) from April to August 2014 at Northeast Normal University. The microcosms were placed in phytotrons under a light intensity of 350 μmol^−2^ S^−1^ (06:00–20:00) and a relative humidity of 40%–60%. The temperature of the microcosms matched the average summer temperature in the Songnen meadow over the last 10 years. The temperature and N input in the phytotrons were based on those described by Zhang et al. [35]. The control and N input treatments were set up as follows: 22 °C from 06:00–10:00, 25 °C from 10:00–15:00, 22 °C from 15:00–20:00, and 22 °C from 20:00–06:00. In the warming and combined warming and N input treatments, the temperature was increased by 3 °C in all time periods relative to the control and N input treatments. The N input and combined warming and N input treatments received N (10 g m^−2^ yr^−1^). The soil water content was maintained at 50%–60% of the field capacity by adding water every two days.

After twelve weeks of growth, we harvested the plants. The aboveground samples were cut at the soil surface, removed from the microcosms, rinsed with deionized water, dried at 65 °C for 48 h, and then weighed. The plant roots were collected and washed using deionized water to measure mycorrhizal colonization. To account for soil heterogeneity, soil was collected from three random locations in each microcosm and was sieved using a 2 mm soil sieve. Dried plant leaf and soil samples were milled in a ball mill prior to C, N, and P chemical analysis.

The total C and N contents in the soil and plant leaves were determined using a stable isotope mass spectrometer (Isoprime 100, Isoprime Ltd, Manchester, UK). Soil and plant leaves were digested in sulfuric acid, and then total P contents were determined photometrically using the molybdenum blue ascorbic acid method [42]. The root samples were cut into 1 cm segments and were cleared with 10% (w/v) KOH and then stained with trypan blue at 90 °C for 2 h. AMF colonization was estimated using a previously described method [35].

### 2.3. Statistical Analysis

All of the data were tested for normality and homogeneity of variance before analysis. All plant and soil data were analyzed using ANOVA with warming, N addition, and benomyl treatment as factors, and their interactions were assessed. In case of significant interactions, means were compared using Tukey’s HSD test. These ANOVAs were followed by an individual *t*-tests to detect differences between the NM and AM treatments. All statistical analyses were performed using SPSS software (SPSS 16.0 for Windows, Chicago, IL, USA).

## 3. Results

### 3.1. Mycorrhizal Colonization

No colonization was observed in the NM treatments. However, in AM treatments, mycorrhizal colonization of the three plant species varied within the same treatment; specifically, the colonization of *S. viridis* and *L. chinensis* was significantly greater than that of *S. corniculata* (*p* < 0.05) except in the treatment with warming and N input combined (Table 1).

### 3.2. Plant Nutrients and Stoichiometric Ratios

Three-way factorial analyses of variance revealed significant main effects of warming on the C, N, and P contents and stoichiometric ratios of the three plant species except for C and P contents in *S. corniculata* and the C:P ratio in *S. viridis* and *S. corniculata*. Significant main effects of N input on the C, N, and P contents of *S. corniculata*; the N content of *S. viridis*; the C:N and N:P ratios of all three species; and the C:P ratio of *L. chinensis* were detected. Significant interaction effects of W × N on the C, N, and P contents of *S. viridis*; the C:N ratio of *L. chinensis*; the C:P ratios of S. viridis and *L. chinensis*; and the N:P ratios of *L. chinensis* and *S. corniculata* were observed (Table 2). 

AMF increased the C content of *S. viridis* by 119% and 138% (*p* < 0.05; Figure 1A) in the N and WN treatments, respectively, and increased the C content of *L.*
*chinensis* by 282% (*P* < 0.01; Figure 1B) in the N treatment. There were significant effects of AMF × N and AMF × W on C content in *S. viridis* and *L. chinensis*, respectively (Table 2), whereas C contents in *S. corniculata* and N content in *S. viridis* were not affected by AMF (Figure 1C,D; Table 2).

AMF increased the N content of *L. chinensis* by 231% (*p* < 0.01; Figure 1E) in the N treatments and decreased the N content of *S. corniculata* by 52% (*p* < 0.001, Figure 1F) in the WN treatments. There were significant effects of AMF ×W and AMF × N on N content in *L. chinensis* and *S. corniculata*, respectively (Table 2).

AMF increased the P content of *S. viridis* by 54%, 118% (*p* < 0.05), and 115% (*p* < 0.05, Figure 1G) in the W, N, and WN treatments, respectively, and by 299% (*p* < 0.01; Figure 1H) in the *L. chinensis* in the N treatment but decreased the P content in *S. corniculata* by 42% (*p* < 0.05, Figure 1I) in the WN. AMF × W and AMF × W × N had significant effects on the P content in *L. chinensis* and *S. corniculata*, respectively (Table 2).

AMF increased the C:N ratio of *S. viridis* by 74% (*p* < 0.01), 57% (*p* < 0.05), 122% (*p* < 0.001), and 121% (*p* < 0.01; Figure 2A) in the control, warming, N input and combination of warming and N input, respectively. AMF decreased the C:N ratio of *L. chinensis* by 36% (*p* < 0.001) in the CK treatment, but in the N and WN treatments, AMF increased the C:N ratio of *L. chinensis* by 17% (*p* < 0.01) and 48% (*p* < 0.001; Figure 2B), respectively. There were significant effects of AMF × W and AMF × N on the C:N ratio of *L. chinensis* (Table 2). AMF significantly increased the C:N ratio of *S. corniculata* by 114% (*p* < 0.001) and 25% (*p* < 0.01; Figure 2C) in the N and WN treatments, respectively.

AMF increased the C:P ratio of *S. viridis* by 11% (*p* < 0.05; Figure 2D) in the WN treatment and that of *S. corniculata* by 18% (*p* < 0.05; Figure 2F) in the CK treatment; it had no impact in the other treatments. There were significant effects of AMF × N on the C:P ratios of *S. viridis* and *L. chinensis* and of AMF × W and AMF × W × N on the C:P ratio of *S. corniculata* (Table 2). 

AMF decreased the N:P ratio of *S. viridis* by 51%, 48% (*p* < 0.01), 51% (*p* < 0.001), and 52% (*p* < 0.01; Figure 2G) in the in the control, warming, N input and combination of warming and N input, respectively. AMF increased the N:P ratio of *L. chinensis* by 46% (*p* < 0.01) in the CK treatment but decreased that of *L. chinensis* by 16% (*p* < 0.05) and 27% (*p* < 0.01; Figure 2H) in the N and WN treatments, respectively. AMF decreased the N:P ratio of *S. corniculata* by 49% (*p* < 0.001) and 15% (*p* < 0.05; Figure 2I) in the N and WN treatments, respectively. There were significant effects of AMF × N on the N:P ratio of all three species and of AMF × W and AMF × W × N on the N:P ratio of *L. chinensis* and *S. corniculata* (Table 2).

### 3.3. Soil Nutrients and Stoichiometric Ratios

Three-way factorial analyses of variance showed significant differences on the effects of warming on soil N and P contents and the C:N ratio; of N input on soil N and P contents and the C:N, C:P and N:P ratios; and of W × N on the C:P ratio (Table 3).

AMF had no effects on soil nutrients and stoichiometric ratios (Table 3); however, under N input, the soil N content and the N:P ratio were reduced by 18% (*p* < 0.05) and 29% (*p* < 0.05; Figure 3B,F), respectively, and the soil C:N ratio was increased by 22% (*p* < 0.05; Figure 3D) in the presence of AMF. In the WN treatment, AMF decreased the soil P content by 22% (*p* < 0.05; Figure 3C) and increased the soil N:P ratio by 44% (*p* < 0.05; Figure 3F). Significant effects of W × AMF on soil N and P contents and the C:N, C:P and N:P ratios; of N × AMF on soil P content and the C:P and N:P ratios; and of W × N × AMF on the soil N:P ratio were observed (Table 3).

## 4. Discussion

### 4.1. Effect of AMF on Plant C:N:P Stoichiometry

Previous studies demonstrated that terrestrial carbon and nitrogen pools can be significantly stimulated by experimental N input [43,44], which could be partly explained by warming-induced increases in net soil N mineralization and nitrification rates [45]. The significant effect of N input on plant C:N and N:P ratios may be attributed to higher soil N availability, which stimulates plant growth [23]. However, our results indicated that AMF alter different plant C:N:P stoichiometries under warming and nitrogen input in a semiarid meadow. 

Our previous study has shown that under warming treatment, the mycorrhizal benefits increased by 374.4% for the aboveground biomass of *S. viridis* [35]. In the present study, under warming treatment, AMF significantly increased the P contents and the C:N ratio of *S. viridis*. AMF may increase plant biomass by promoting nutrient cycling [31].

In the N input treatment of our previous study [35], the mycorrhizal benefits conferred to the aboveground biomass of *S. viridis* and *L. chinensis* were increased by 51.1% and 47.4%, respectively, whereas the aboveground biomass of *S. corniculata* decreased significantly under both treatments. These results are generally in agreement with those of the present study. In the present study, under N input and a combination of warming and N input, AMF significantly enhanced the C and P contents and the C:N ratio of the C_4_ grass *S. viridis*. Under N input, AMF significantly increased the C, N, and P contents and the C:N ratio of the C_3_ grass *L. chinensis.* However, AMF significantly decreased the C, N, and P contents of *S. corniculata* under the combination of warming and N input. The mycorrhizal benefits conferred to C_4_ grasses are greater than those conferred to C_3_ grasses and should therefore result in higher carbon production and increased AMF development [41,46], an expectation consistent with the findings of the current study. However, in *S. corniculata*, AMF symbiosis was antagonistic. The results suggest that the contribution of AMF to the three dominant species in the Songnen meadow steppe varies under global change. These results support our hypothesis and further suggest that plant stoichiometric responses to global change and ecosystem stability can be adjusted by AMF.

N and P are the most common limiting elements for plant growth and have profound impacts on plant functions [30]. According to some studies, AMF can transfer a considerable amount of N from the soil to host plants [47,48], but other studies have found no evidence that AMF symbioses increase N uptake [49,50]. Phosphorus is the most readily immobilized element in the soil, and its availability is very low [51]; therefore, mycorrhizal P uptake is the dominant pathway [52]. AMF form extensive hyphal networks in the soil and forage effectively for nutrients, especially P, which is supplied to their host plants [10,53]. Fungal nutrient allocation is adjusted through the carbon source strength of individual host plants [54], and plant species affect the AMF response to resource stoichiometry [55].

Furthermore, under all treatments, AMF significantly decreased the N:P ratio of *S. viridis* under N input and a combination of warming and N input, and AMF significantly decreased the N:P ratios of *L. chinensis* and *S. corniculata*. Several studies have suggested that a plant N:P ratio < 14 indicates N limitation and that a plant N:P ratio > 16 indicates P limitation [56,57]. Numerous studies have shown that N input induces an imbalance in the N:P ratio and an increase in P limitation in grasslands [58,59]. In the present study, N input and the combination of warming and N input significantly enhanced the plant N:P ratios of the three species, which may result in an altered balance between N and P. Therefore, the Songnen meadow steppe ecosystem changed from being N limited to being P limited, which is in accordance with an experiment in a temperate steppe ecosystem [60]. However, AMF significantly reduced the N:P ratios of the three species under N input and the combination of warming and N input, which agrees with a previous result [61]. The trade balance model predicts that N enrichment of a P-limited soil will exacerbate the P limitation and increase the amount of P obtained through symbiosis [30]; thus, the plant N:P ratio will decrease in the presence of AMF. The results suggest that AMF might slow the increase in P limitation caused by global change in Songnen meadows. Rational management of soil nutrients in these meadows is critically important to increase plant productivity and to improve the sustainable utilization of grassland ecosystems.

### 4.2. Effects of AMF on Soil C:N:P Stoichiometry

Stoichiometry is a vital indicator of biogeochemical cycles in terrestrial ecosystems [62]. Soil C:N:P stoichiometry provides a crucial potential diagnostic value for nutrient mineralization and organic matter decomposition [63]. Studies have revealed that C:N:P stoichiometry in soil and plants is tightly linked [64,65]. Soil C:N:P stoichiometry not only regulates microbial activity but also plant N and P uptake [64], while plant C:N:P stoichiometry directly reflects the availability of soil N and P. Many studies have suggested that AMF may transport large numbers of limiting nutrients (N, P) to their host plants from the soil [48,66] and may promote nutrient use efficiency by accelerating the decomposition of organic matter [48,67].

Yue et al. [23] found that high stoichiometric homeostasis, measured as the soil C:N ratio, decreased significantly under N addition. In the present study, under N input, the soil N content and the N:P ratio were decreased significantly in the presence of AMF, but the soil C:N ratio under N input and AMF increased, suggesting that AMF play vital roles in soil nutrient cycling [68], including C and N cycling, in grassland ecosystems [45]. The results indicate that AMF might improve the soil stability and sustainability of plant-soil systems.

## 5. Conclusions

While N is proposed as the primary limiting nutrient for plant growth, warming and N input resulted in a change from N limitation to P limitation in a semiarid meadow steppe; therefore, P was arguably more limiting in the grassland ecosystem in the study area. Our results showed that AMF play a vital role in maintaining plant nutrient balance and affect plant growth by altering plant nutrient uptake. AMF might slow the rate of P limitation in the Songnen meadow steppe under global change, which would alter plant community composition under future global change. In addition, AMF enhance the soil stability and sustainability of plant-soil systems.

## Figures and Tables

**Figure 1 ijerph-16-00397-f001:**
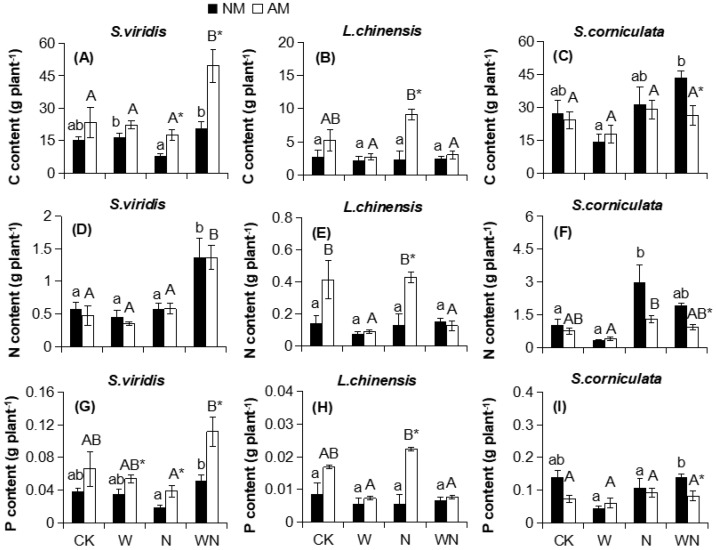
Effects of AMF on the C, N, and P contents of *S. viridis* (**A**,**D**,**G**), *L. chinensis* (**B**,**E**,**H**), and *S. corniculata* (**C**,**F**,**I**) under warming and N input. Different lowercase and capital letters above bars indicate significant differences (*p* < 0.05) among different warming and N input treatments within the same AMF treatment. Asterisks indicate significant differences (*p* < 0.05) between AMF treatments within the same warming and/or N input condition. CK, control; N, N input; W, warming; and WN, combination of warming and N input.

**Figure 2 ijerph-16-00397-f002:**
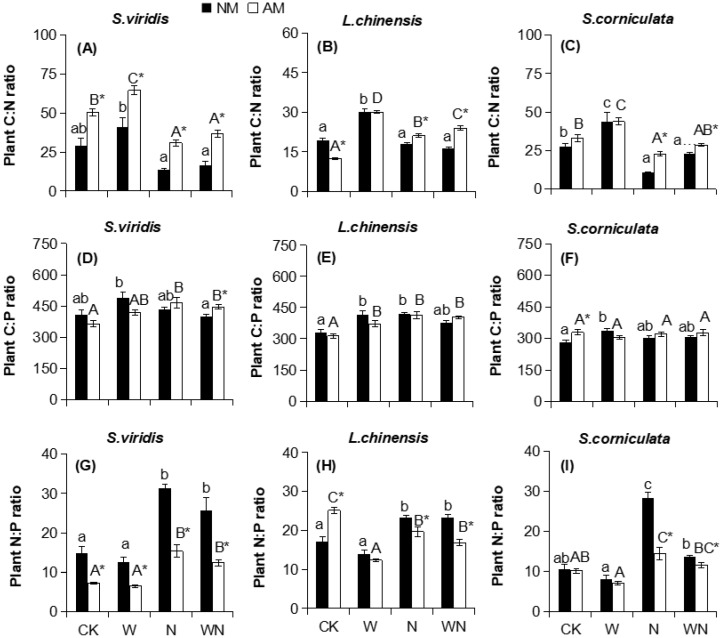
Effects of AMF on the C:N, C:P, and N:P ratios of *S. viridis* (**A**,**D**,**G**, *L. chinensis* (**B**,**E**,**H**) and *S. corniculata* (**C**,**F**,**I**) under warming and N input. Different lowercase and capital letters above bars indicate significant differences (*p* < 0.05) among different warming and N input treatments within the same AMF treatment. Asterisks indicate significant differences (*p* < 0.05) between AMF treatments within the same warming and/or N input condition. CK, control; N, N input; W, warming; and WN, combination of warming and N input.

**Figure 3 ijerph-16-00397-f003:**
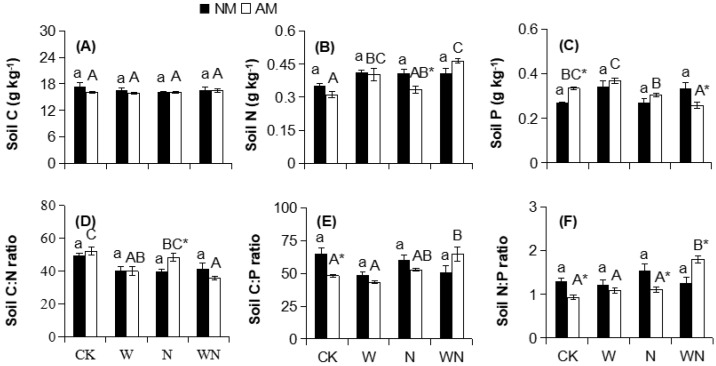
Effects of AMF on soil C (**A**), N (**B**), and P (**C**) contents and C:N (**D**), C:P (**E**), and N:P (**F**) ratios under warming and N input. Different lowercase and capital letters above bars indicate significant differences (*p* < 0.05) among different warming and N input treatments within the same AMF treatment. Asterisks indicate significant differences (*p* < 0.05) between AMF treatments within the same warming and/or N input condition. CK, control; N, N input; and W, warming; WN, combination of warming and N input.

**Table 1 ijerph-16-00397-t001:** The colonization status (%) of arbuscular mycorrhizal fungi (AMF) treatment under warming (W) and N input (N). CK, control; N, N input; W, warming; W × N, combination of warming and N input.

Treatment	*S. viridis*	*L. chinensis*	*S. corniculata*
CK	83.33 ± 10.00 ^b^	68.34 ± 1.67 ^b^	33.34 ± 3.34 ^a^
W	88.33 ± 5.00 ^b^	83.33 ± 5.00 ^b^	16.67 ± 3.34 ^a^
N	65.00 ± 8.33 ^b^	77.26 ± 0.59 ^b^	38.33 ± 5.00 ^a^
W × N	50.00 ± 6.67 ^a^	53.50 ± 16.83 ^a^	45.61 ± 10.00 ^a^

Note: The colonization status with different superscripts (a or b) differ significantly at *p* = 0.05 among different plant species in the same treatment.

**Table 2 ijerph-16-00397-t002:** Results of three-way factorial analyses of variance (ANOVAs) of the effects of warming, N input and arbuscular mycorrhizal fungi (AMF) on plant C, N, and P contents and ratios of C:N, C:P, and N:P.

Plant	Variable	W	N	W × N	AMF	AMF × W	AMF × N	AMF × W × N
*S. viridis*	C	15.38 **	2.583	15.32 **	20.79 ***	2.19	4.65 *	3.41
*L. chinensis*		12.25 **	2.43	1.29	15.55 **	10.18 **	2.69	2.67
*S. corniculata*		0.50	11.43 **	4.15	1.92	0.34	2.11	2.33
*S. viridis*	N	9.76 **	22.99 ***	18.47 ***	0.207	0.01	0.22	0.01
*L. chinensis*		19.47 ***	0.68	0.49	13.36 **	14.50 **	0.02	0.20
*S. corniculata*		8.01 **	26.41 ***	0.19	10.91 **	1.54	7.95 **	0.15
*S. viridis*	P	8.74 **	0.81	15.80 **	16.94 ***	0.99	1.21	2.47
*L. chinensis*		10.89 **	0.23	0.02	11.77 **	7.85 **	0.96	1.40
*S. corniculata*		0.80	10.76 **	3.97	2.93	0.02	2.28	3.34 **
*S. viridis*	C:N	13.78 **	88.52 ***	3.713	77.74 ***	0.29	0.7	0.01
*L. chinensis*		212.46 ***	38.73 ***	183.63 ***	3.97	33.64 ***	77.07 ***	1.41
*S. corniculata*		33.57 ***	64.74 ***	1.41	9.31**	2.34	2.43	0.01
*S. viridis*	C:P	2.32	1.755	13.28 **	0.501	0.08	13.69 **	0.69
*L. chinensis*		5.48 *	22.07 ***	26.50 ***	1.01	0.03	4.55 *	2.29
*S. corniculata*		2.18	0.04	0.34	3.04	6.45*	0.35	6.59 *
*S. viridis*	N:P	6.77 *	91.98 ***	1.49	88.19 ***	0.89	12.03 **	0.05
*L. chinensis*		51.63 ***	29.90 ***	25.06 ***	1.85	21.72 ***	38.42 ***	6.92 *
*S. corniculata*		70.84 ***	134.85 ***	19.23 ***	40.67 ***	16.85 ***	28.08 ***	20.59 ***

* represents a significant difference at *p* < 0.05; ** represents a significant difference at *p* < 0.01; *** represents a significant difference at *p* < 0.001.

**Table 3 ijerph-16-00397-t003:** Results of three-way factorial analyses of variance (ANOVAs) of the effects of warming, N input and arbuscular mycorrhizal fungi (AMF) on soil total C, N, and P contents and ratios of C:N, C:P, and N:P.

Variable	W	N	W × N	AMF	AMF × W	AMF × N	AMF × W × N
Soil C	0.01	0.20	2.13	2.37	0.07	1.55	0.15
Soil N	33.17 ***	8.50 **	0.22	1.99	10.80 **	0.32	4.04
Soil P	7.88 **	11.05 **	4.04	1.21	11.20 **	8.37 **	2.29
Soil C:N	23.62 ***	6.99 *	2.39	0.66	7.20 *	0.02	2.72
Soil C:P	3.57	6.16 *	5.88*	2.60	10.81 **	8.32 **	1.17
Soil N:P	3.05	17.92 ***	1.58	2.05	20.15 ***	4.95 *	7.35 *

* represents a significant difference at *p* < 0.05; ** represents a significant difference at *p* < 0.01; *** represents a significant difference at *p* < 0.001.

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
