# Peer review of "Arbuscular Mycorrhizal Fungi Alter Plant and Soil C:N:P Stoichiometries Under Warming and Nitrogen Input in a Semiarid Meadow of China"

_ijerph, 2019, doi:10.3390/ijerph16030397_

Reviewer 1 Report

section 2.1, 2.2. and 2.3 can be combined

likewise  there is no sense of making two different section for experimental details as given in 2.4 (labelled as 2.3) and 2.5

Legends given in Table 1 is very confusing and not reader friendly. so data given in Table 1 is hard to follow

Again legends in Table 1 and 2 are not consistent, is it W+N or WxN ?

Discussion part 4.1 - could be subdivided in section with stepwise observations made

Detailed discussion is required as many points promised in M&M are not given

Author Response

     Thank you for your comments and suggestion on our manuscript. We have studied the comments carefully and have made corrections, which we hope will meet your approval. To improve the English, we asked a professional English editor from American Journal Experts to correct our manuscript. The following is the explanation of how we complied with the reviewers’ suggestions. Attached please find the revised version, which we would like to submit for your kind consideration.

Point 1: Section 2.1, 2.2. and 2.3 can be combined. 

Response 1: We combined Sections 2.1, 2.2 and 2.3.

Point 2: Likewise  there is no sense of making two different section for experimental details as given in 2.4 (labelled as 2.3) and 2.5.

Response 2: We combined Sections 2.4 and 2.5.

Point 3: Legends given in Table 1 is very confusing and not reader friendly. so data given in Table 1 is hard to follow.

Response 3: We revised the legend for Table 1. We also revised the heading of Section 3.1 to "Mycorrhizal colonization” in order to express the meaning more clearly.

Point 4: Again legends in Table 1 and 2 are not consistent, is it W+N or W × N.

Response 4: We unified the legends for Tables 1 and 2. W × N is correct.

Point 5: Discussion part 4.1 - could be subdivided in section with stepwise observations made. Detailed discussion is required as many points promised in M&M are not given.

Response 4: We revised Section 4.1. We subdivided the section to include stepwise observations. First, we discussed the effect of warming and N input on plant C:N:P stoichiometry. Second, combined with a discussion of our previous study, we discussed the effects of AMF on plant C, N, P and C:N, C:P ratios under different treatments. Finally, we discussed the effects of AMF on plant N:P ratios and added more detailed discussion.

Reviewer 2 Report

Title should read: Arbuscular mycorrhizal fungi alter plant and soil C:N:P stoichiometries under warming and nitrogen input in a semiarid meadow of China

line 31: add Global climate change

line 163: should read: .....followed by 'an' individual

line 176: should read: Note, Different lower case letters ...... 

line 263: should read: .......... showed significant differences on the effects of warming on ....

line 297 on to line 298: re-write this line.

line 310: should read: our previous study has shown that ........

line 320: remove 'original'

line 366: ....., P was arguably more limiting in grassland ecosystem in the area that was studied.

Author Response

Thank you for your comments and suggestion on our manuscript. We have studied the comments carefully and have made corrections, which we hope will meet your approval. To improve the English, we asked a professional English editor from American Journal Experts to correct our manuscript. The following is the explanation of how we complied with the reviewers’ suggestions. Attached please find the revised version, which we would like to submit for your kind consideration.

Point 1: Title should read: Arbuscular mycorrhizal fungi alter plant and soil C:N:P stoichiometries under warming and nitrogen input in a semiarid meadow of China.

Response 1: We revised the title to "Arbuscular mycorrhizal fungi alter plant and soil C:N:P stoichiometries under warming and nitrogen input in a semiarid meadow of China."

Point 2: line 31: add Global climate change.

Response 2: We added global climate change.

Point 3: line 163: should read: .....followed by 'an' individual.

Response 3: We revised it as suggested.

Point 4: line 176: should read: Note, Different lower case letters ......

Response 4: We revised it as suggested.

Point 5: line 263: should read: .......... showed significant differences on the effects of warming on ....

Response 5: We revised it as suggested.

Point 6: line 297 on to line 298: re-write this line.

Response 6: We rewrote this line as follows: “AMF alter different plant C:N:P stoichiometries under warming and nitrogen input in a semiarid meadow.”

Point 7: line 310: should read: our previous study has shown that ........

Response 7: We revised it as suggested.

Point 8: line 320: remove 'original'

Response 8: We removed 'original.'

Point 9: line 366: ....., P was arguably more limiting in grassland ecosystem in the area that was studied.

Response 9: We revised this sentence as follows: ....., P was arguably more limiting in grassland ecosystem in the area that was studied."